# Entropy Optimization, Maxwell–Boltzmann, and Rayleigh Distributions

**DOI:** 10.3390/e23060754

**Published:** 2021-06-15

**Authors:** Nicy Sebastian, Arak M. Mathai, Hans J. Haubold

**Affiliations:** 1Department of Statistics, St. Thomas College, Thrissur, Kerala 680001, India; nicycms@gmail.com; 2Department of Mathematics and Statistics, McGill University, Montreal, QC H0H H9X, Canada; a.mathai@mcgill.ca; 3Office for Outer Space Affairs, United Nations, Vienna International Center, A-1400 Vienna, Austria

**Keywords:** complex Maxwell–Boltzmann and Rayleigh densities, multivariate and matrix-variate densities, matrix-variate pathway models, type-1, type-2 beta densities, generalized gamma, generalized entropy, optimization of entropy, ellipsoid of concentration, 62E15, 62E10, 62B10, 60B20, 49K45, 15A15, 15B52

## Abstract

In physics, communication theory, engineering, statistics, and other areas, one of the methods of deriving distributions is the optimization of an appropriate measure of entropy under relevant constraints. In this paper, it is shown that by optimizing a measure of entropy introduced by the second author, one can derive densities of univariate, multivariate, and matrix-variate distributions in the real, as well as complex, domain. Several such scalar, multivariate, and matrix-variate distributions are derived. These include multivariate and matrix-variate Maxwell–Boltzmann and Rayleigh densities in the real and complex domains, multivariate Student-t, Cauchy, matrix-variate type-1 beta, type-2 beta, and gamma densities and their generalizations.

## 1. Introduction

The following notations will be used in this paper: Real scalar variables, whether mathematical variables or random variables, will be denoted by lower-case letters, such as x,y, etc.; real vector/matrix variables—mathematical and random—will be denoted by capital letters, such as X,Y, etc. Complex variables will be written with a tilde, such as $\tilde{x},\tilde{y},\tilde{X},\tilde{Y}$, etc. Scalar constants will be denoted by a,b, etc., and vector/matrix constants by *A*,*B*, etc. No tilde will be used on constants. If $A=(a_{ij})$ is a p×p matrix, then its determinant will be denoted by |A| or det(A) if the elements of $a_{ij}$ are real or complex. The transpose of *A* is written as $A^{\prime }$ and the complex conjugate transpose as $A^{*}$. The absolute value of the determinant will be written as $\left|\det\left(A\right)\right|=\sqrt{\det(AA^{*})}$. For example, if $\det\left(A\right)=a+ib,i=\sqrt{\left(-1\right)},a,b$ is a real scalar, then the absolute value is $\left|\det\left(A\right)\right|=\sqrt{\left(a^{2}+b^{2}\right)}$. If $X=(x_{ij})$ is a p×q real matrix, then the wedge product of the differentials $dx_{ij}$ is written as $dX=\wedge_{i=1}^{p}\wedge_{j=1}^{q}dx_{ij}$, where, for two real scalar variables *x* and *y* with differentials dx and dy, the wedge product is defined as dx∧dy=−dy∧dx so that dx∧dx=0,dy∧dy=0. If $\tilde{X}$ in the complex domain is a p×q matrix, then we can write $\tilde{X}=X_{1}+iX_{2},i=\sqrt{\left(-1\right)},X_{1},X_{2}$, which is real; then, we define $d\tilde{X}=dX_{1}\wedge dX_{2}$. If f(X) is a real-valued scalar function of *X*, where *X* may be scalar real variable *x*, scalar complex variable $\tilde{x}$, vector/matrix real variable *X*, or vector/matrix complex variable $\tilde{X}$ such that f(X)≥0 for all *X* and $\int_{X}f\left(X\right)dX=1$, then f(X) will be called a statistical density.

In many disciplines, especially in physics, communication theory, engineering, and statistics, one popular method of deriving statistical distributions is the optimization of an appropriate measure of entropy under appropriate constraints. For a real scalar random variable *x*, [1] introduced a measure of entropy or a measure of uncertainty: (1)$$S\left(f\right)=-c\int_{x}f\left(x\right)\ln f\left(x\right)dx\;$$
where *c* is a constant. The corresponding measure for the discrete case is
$$-c\sum_{j=1}^{k}p_{j}\ln p_{j},p_{j}>0,j=1,\ldots ,k,p_{1}+\ldots +p_{k}=1$$
or $\left(p_{1},\ldots ,p_{k}\right)$, which is a discrete probability law. By optimizing S(f), several authors have derived exponential, Gaussian, and other distributions under the constraints in terms of moments of *x*, such as E[x]= fixed over all functional *f*, meaning that the first moment is given, where E(·) indicates the expected value of (·). This constraint will produce exponential density. If E[x] and $E[x^{2}]$ are fixed, meaning that the first two moments are fixed, then one has a Gaussian density, etc. The basic entropy measure in (1) has been generalized by various authors. One such generalized entropy is the Havrda–Charvát entropy [2] $H_{\alpha }\left(f\right)$ for the real scalar variable *x*, which is given by
(2)$$H_{\alpha }\left(f\right)=\frac{\int_{x}{\left[f\left(x\right)\right]}^{\alpha }dx-1}{2^{1-\alpha }-1},\alpha \neq 1\;$$
where f(x) is a density. The original $H_{\alpha }\left(f\right)$ is for the discrete case, and the corresponding continuous case is given in (2). Various properties, characterizations, and applications of the Shannon entropy and various α-generalized entropies were discussed by [3]. A modified version of (2) was introduced by Tsallis [4], and it is known in the literature as Tsallis’ entropy, which is the following:(3)$$T_{q}\left(f\right)=\frac{\int_{x}{\left[f\left(x\right)\right]}^{q}dx-1}{1-q},q\neq 1.\;$$
Observe that when α→1 in (2) and q→1 in (3), both of these generalized entropies in the real scalar case reduce to the Shannon entropy of (1). Tsallis developed the whole area of non-extensive statistical mechanics by deriving Tsallis’ statistics by optimizing (3) under the constraint that the first moment is fixed in an escort density, $g\left(x\right)=\frac{{\left[f\left(x\right)\right]}^{q}}{\int_{x}{\left[f\left(x\right)\right]}^{q}dx}$. Hundreds of papers have been published on Tsallis’ statistics.

In early 2000, the second author introduced a generalized entropy of the following form:(4)$$M_{\alpha }\left(f\right)=\frac{\int_{X}{\left[f\left(X\right)\right]}^{1+\frac{a-\alpha }{\eta }}dX-1}{\alpha -a},\alpha \neq a\;$$
where f(X) is a statistical density, $f\left(X\right)\geq 0,\int_{X}f\left(X\right)dX=1$, where *X* may be real scalar *x*, complex scalar $\tilde{x}$, real vector/matrix *X*, or complex vector/matrix $\tilde{X}$, *a* is a fixed real scalar anchoring point, α is a real scalar parameter, and η>0 is a real scalar constant so that the deviation of α from *a* is measured in η units. In the real scalar case, we can see that when α→a, then (4) goes to the Shannon entropy in (1). Therefore, for vector/matrix variables in the real and complex domain, one has a generalization of the Shannon entropy in (4). If (3) is optimized under the constraint that the first moment E[x] in f(x) is fixed; then, it does not lead directly to Tsallis’ statistics. One must optimize (3) in the escort density mentioned above under the restriction that the first moment in the escort density is fixed. Then, one obtains Tsallis’ statistics. If (4) is used, then one can derive various real and complex, scalar, vector, or matrix-variate distributions directly from f(X) by imposing moment-like restrictions in f(X). A particular case of (4) for a=1,η=1, introduced by the second author was applied by [5] in time-series analysis, fractional calculus, and other areas. The researchers in [6] used a particular case of (4) in record values, ordered random variables, and derived some properties, including characterization theorems. In [7] discussed the analytical properties of the classical Mittag–Leffler function as being derived as the solution of the simplest fractional differential equation governing relaxation processes. In [8] studied the complexity of the ultraslow diffusion process using both the classical Shannon entropy and its general case with the inverse Mittag–Leffler function in conjunction with the structural derivative.

In the present article, the term “entropy” is used as a mathematical measure of uncertainty or information characterized by some basic axioms, as illustrated by [3]. Thus, it is a functional resulting from a set of axioms, that is, a function that can be interpreted in terms of a statistical density in the continuous case and in terms of multinomial probabilities in the discrete case. A general discussion of “entropy” is not attempted here because, as per Von Neumann, “whoever uses the term ‘entropy’ in a discussion always wins since no one knows what entropy really is, so in a debate, one always has the advantage”. An overview of various entropic functional forms used so far in the literature is available from [9], along with their historical backgrounds and an account of the numbers of citations of these various functional forms. Hence, no detailed discussion of various entropic functional forms is attempted in the present paper. The concept of entropy is applied in general physics, information theory, chaos theory, time series, computer science, data mining, statistics, engineering, mathematical linguistics, stochastic processes, etc. An account of the entropic universe was given by [10], along with answers to the following questions: How different concepts of entropy arose, what the mathematical definitions of each entropy are, how entropies are related to each other, which entropy is appropriate in which areas of application, and their impacts on the scientific community. Hence, the present article does not attempt to repeat the answers to these questions again. The present paper is about one entropy measure on a real scalar variable, its generalizations to vector/matrix variables in the real and complex domains, and an illustration of how this entropy can be optimized under various constraints to derive various statistical densities in the scalar, vector, and matrix variables in the real and complex domains. Because the entropy measure to be considered in the present article does not contain derivatives, the method of calculus of variation is used for optimization so that the resulting Euler equations will be simple. Mathematical variables and random variables are treated in the same way so that the double notations used for random variables are avoided. In order to avoid having too many symbols and the resulting confusion, scalar variables are denoted by lower-case letters and vector/matrix variables are denoted by capital letters so that the presentation is concise, consistent, and reader-friendly.

### Entropy as an Expected Value

Shannon entropy S(f) can be looked upon as an expected value of −clnf(x). In Mathai’s entropy (4), one can write the numerator as $\int_{X}\left\{{\left[f\left(X\right)\right]}^{\frac{a-\alpha }{\eta }}-1\right\}f\left(X\right)dX$, which is the expected value of ${\left[f\left(X\right)\right]}^{\frac{a-\alpha }{\eta }}-1$. Then, (4) is the following expected value:(5)$$M_{\alpha }\left(f\right)=E\left[\frac{{\left\{f\left(X\right)\right\}}^{\frac{a-\alpha }{\eta }}-1}{\alpha -a}\right].\;$$
The quantity in the expected value operator goes to $-\frac{1}{\eta }\ln f\left(X\right)$ when α→a, which is the same as the Shannon case for $c=\frac{1}{\eta }$. Therefore, the quantity inside the expectation operator is an approximation to $-\frac{1}{\eta }\ln f\left(X\right)$.

## 2. Optimization of Mathai’s Entropy for the Real Scalar Case

Let *x* be a real scalar variable and let f(x) be a density function, that is, f(x)≥0 for all *x* and $\int_{x}f\left(x\right)dx=1$. Consider the optimization of (4) under the following moment-like constraints:$$E\left[x^{\gamma (\frac{a-\alpha }{\eta })}\right]=\mathrm{fixed}\;\mathrm{and}\;E\left[x^{\gamma (\frac{a-\alpha }{\eta })+\delta }\right]=\mathrm{fixed}$$
over all possible densities f(x). Then, if we use calculus of variation for the optimization of (4), the Euler equation is the following:
(6)$$\begin{array}{cc} \frac{\partial }{\partial f}[f^{1+(\frac{a-\alpha }{\eta })} & -\lambda_{1}x^{\gamma (\frac{a-\alpha }{\eta })}f\left(x\right)+\lambda_{2}x^{\gamma (\frac{a-\alpha }{\eta })+\delta }f\left(x\right)]=0\Rightarrow \\ \left(1+\frac{a-\alpha }{\eta }\right)f^{\frac{a-\alpha }{\eta }} & =\lambda_{1}x^{\gamma (\frac{a-\alpha }{\eta })}\left[1-b\left(a-\alpha \right)x^{\delta }\right]\Rightarrow \\ f_{1}\left(x\right) & =c_{1}x^{\gamma }{\left[1-b\left(a-\alpha \right)x^{\delta }\right]}^{\frac{\eta }{a-\alpha }},\alpha <a \end{array}$$
where $\lambda_{1}$ and $\lambda_{2}$ are Lagrangian multipliers and $\frac{\lambda_{2}}{\lambda_{1}}$ is taken as b(a−α) for convenience for α<a,b>0,γ>0,δ>0,η>0; *a* is a fixed real scalar constant, $1-b\left(a-\alpha \right)x^{\delta }>0$, and $c_{1}$ is the normalizing constant. For α>a, $f_{1}\left(x\right)$ changes into
(7)$$f_{2}\left(x\right)=c_{2}x^{\gamma }{\left[1+b\left(\alpha -a\right)x^{\delta }\right]}^{-\frac{\eta }{\alpha -a}}\;$$
for α>a, b>0,η>0,δ>0,γ>0,x≥0. When α→a, both $f_{1}\left(x\right)$ and $f_{2}\left(x\right)$ go to
(8)$$f_{3}\left(x\right)=c_{3}x^{\gamma }e^{-b\eta x^{\delta }}\;$$
for b>0,η>0,δ>0,γ>0,x≥0. Observe that all three functions $f_{i}\left(x\right),i=1,2,3$ can be reached through the pathway parameter α. From $f_{1}\left(x\right)$, one can go to $f_{2}\left(x\right)$ and $f_{3}\left(x\right)$. Similarly, from $f_{2}\left(x\right)$, one can obtain $f_{1}\left(x\right)$ and $f_{3}\left(x\right)$. Hence, $f_{1}\left(x\right)$ or $f_{2}\left(x\right)$ is Mathai’s pathway model for the real scalar positive variable *x* as a mathematical model or as a statistical model. The model $f_{1}\left(x\right)$ is a generalized type-1 beta model, $f_{2}\left(x\right)$ is a generalized type-2 beta model, and $f_{3}\left(x\right)$ is a generalized gamma model. For δ=2,γ=0, $f_{3}\left(x\right)$ is a real scalar Gaussian model. For γ=2,δ=2, $f_{3}\left(x\right)$ is a Maxwell–Boltzmann density for x≥0, and for $\gamma =\frac{1}{2},\delta =2,x\geq 0$, $f_{3}\left(x\right)$ is the Rayleigh density for the real scalar positive variable case. If a location parameter is desired, then *x* is replaced by x−m in all of the above models, where *m* is the relocation parameter. For γ=0,δ=1,η=1,a=1,α=q, $f_{i}\left(x\right),i=1,2,3$ is Tsallis’ statistic of non-extensive statistical mechanics; see [4] Tsallis (1988). Hundreds of articles have been published on Tsallis’ statistics. For δ=1,η=1,a=1,α=1, $f_{2}\left(x\right)$ and $f_{3}\left(x\right)$—but not $f_{1}\left(x\right)$—provide superstatistics of statistical mechanics. Several articles have been published on superstatistics.

Fermi–Dirac and Bose–Einstein densities are also available from the same procedure. In this case, the second factor $x^{\delta }$ in the constraint is replaced by $e^{cx},c>0,x\geq 0$, and the Lagrangian multipliers are taken as $-\lambda_{1}$ and $-\lambda_{2}$ so that the second factor in Equation (Equation 6) becomes ${\left(\lambda_{1}+\lambda_{2}e^{cx}\right)}^{-\eta /(\alpha -a)}$ for α>a with $(\lambda_{1}+\lambda_{2}e^{cx})>0$ to create a density function. Now, take γ=0,η=1,α−a=1. Then, for $\lambda_{1}=1,\lambda_{2}=e^{d}$ and for some constant *d*, this gives the Fermi–Dirac density, and for $\lambda_{1}=-1$ and $\lambda_{2}=e^{d}$, this gives Bose–Einstein density.

In model-building situations, if $f_{3}\left(x\right)$ is the generalized gamma model, Maxwell–Botlzmann model (γ=2,δ=2), Rayleigh model $\left(\gamma =\frac{1}{2},\delta =2\right)$, or Gaussian model (γ=0,δ=2) and is the stable or ideal situation in a physical system, then $f_{1}\left(x\right)$ and $f_{2}\left(x\right)$ provide the unstable or chaotic neighborhoods, and through the pathway parameter α, one can model the stable situation, the unstable neighborhoods, and the transitional stages in a data analysis situation. This is the pathway idea of Mathai.

## 3. Constraints in Terms of the Ellipsoid of Concentration in the Real p-Variate Case

Let *X* be a p×1 real vector with distinct real scalar variables $x_{j}$ as elements; $X^{\prime }=\left(x_{1},\ldots ,x_{p}\right)$, where a prime denotes the transpose. Let μ be a p×1 location vector. Let the covariance matrix in *X* be $\Sigma =E[\left(X-\mu \right){\left(X-\mu \right)}^{\prime }],\mu =E\left[X\right]$; then, $\Sigma =\Sigma^{\prime }$, and let Σ>O (real positive definite). Then, the square of the Euclidean distance of *X* from the point of location μ is ${\left(X-\mu \right)}^{\prime }\left(X-\mu \right)$, and the generalized distance of *X* from μ is ${\left(X-\mu \right)}^{\prime }\Sigma^{-1}\left(X-\mu \right)$. Because Σ is real positive definite, $u={\left(X-\mu \right)}^{\prime }\Sigma^{-1}\left(X-\mu \right)$ is known as the ellipsoid of concentration. The probability content of this ellipsoid of concentration is an important quantity in statistical analysis. Let us consider constraints in terms of moments of the ellipsoid of concentration *u*. Consider the following constraints:$$E\left[u^{\gamma (\frac{a-\alpha }{\eta })}\right]=\mathrm{fixed}\;\mathrm{and}\;E\left[u^{\gamma (\frac{a-\alpha }{\eta })+\delta }\right]=\mathrm{fixed}$$
over all possible densities f(X), where *X* is a p×1 vector random variable. Then, optimizing Mathai’s entropy in (4) for all possible densities f(X) and proceeding as in Section 2, we have the following three densities: For α<a,
(9)$$f_{1}\left(X\right)=C_{1}{\left[{\left(X-\mu \right)}^{\prime }\Sigma^{-1}\left(X-\mu \right)\right]}^{\gamma }{\left[1-b\left(a-\alpha \right){\left({\left(X-\mu \right)}^{\prime }\Sigma^{-1}\left(X-\mu \right)\right)}^{\delta }\right]}^{\frac{\eta }{a-\alpha }}\;$$
for α<a,b>0,γ>0,δ>0,Σ>O,η>0. For α>a, the model in (9) changes into the model
(10)$$f_{2}\left(X\right)=C_{2}{\left[{\left(X-\mu \right)}^{\prime }\Sigma^{-1}\left(X-\mu \right)\right]}^{\gamma }{\left[1+b\left(\alpha -a\right){\left({\left(X-\mu \right)}^{\prime }\Sigma^{-1}\left(X-\mu \right)\right)}^{\delta }\right]}^{-\frac{\eta }{\alpha -a}}\;$$
for α>a,b>0,γ>0,δ>0,η>0,Σ>O, and for α→a, the models in both (9) and (10) go to the model
(11)$$f_{3}\left(X\right)=C_{3}{\left[{\left(X-\mu \right)}^{\prime }\Sigma^{-1}\left(X-\mu \right)\right]}^{\gamma }e^{-b\eta {\left({\left(X-\mu \right)}^{\prime }\Sigma^{-1}\left(X-\mu \right)\right)}^{\delta }}\;$$
for b>0,η>0,Σ>O, where $C_{i},i=1,2,3$ are the normalizing constants. These normalizing constants can be evaluated, and further properties of the models can be studied with the help of the following results from [11]:

**Lemma** **1.**
*Let $X=(x_{ij})$ be a p×q real matrix with distinct real scalar variables $x_{ij}$ as elements. Let A be p×p and B be q×q constant nonsingular matrices. Then,*
(12)$$Y=AXB,|A|\neq 0,|B|\neq 0\Rightarrow dY={\left|A\right|}^{q}{\left|B\right|}^{p}dX.\;$$


For the proof of this result, as well as for other similar results, see [11]. We will state one more result from [11] here without proof.

**Lemma** **2.***Let X be real p×q,p≤q, and rank p matrix with distinct real scalar variables as elements. Let $S=XX^{\prime }$ so that S is p×p symmetric and positive definite. Then, after integrating out over the Stiefel manifold,*(13)$$dX=\frac{\pi^{\frac{pq}{2}}}{\Gamma_{p}\left(\frac{q}{2}\right)}{\left|S\right|}^{\frac{q}{2}-\frac{p+1}{2}}dS\;$$*where, for example, $\Gamma_{p}\left(\alpha \right)$ is the real matrix-variate gamma given by*(14)$$\Gamma_{p}\left(\alpha \right)=\pi^{\frac{p(p-1)}{4}}\Gamma \left(\alpha \right)\Gamma \left(\alpha -\frac{1}{2}\right)...\Gamma \left(\alpha -\frac{p-1}{2}\right),ℜ\left(\alpha \right)>\frac{p-1}{2}$$(15)$$=\int_{S>O}{\left|S\right|}^{\alpha -\frac{p+1}{2}}e^{-\mathrm{tr}(S)}dS,ℜ\left(\alpha \right)>\frac{p-1}{2}$$*where ℜ(·) indicates the real part of (·), S>O is p×p real positive definite, and tr(·) indicates the trace of (·)*.

### Evaluation of the Normalizing Constants

Consider $f_{1}\left(X\right)$ of (9). Let $Y=\Sigma^{-\frac{1}{2}}\left(X-\mu \right)\Rightarrow dY={\left|\Sigma \right|}^{-\frac{1}{2}}dX$ by using Lemma 1, where $\Sigma^{-\frac{1}{2}}$ is the positive definite square root of $\Sigma^{-1}$. Let $s=Y^{\prime }Y$, which is 1×1 because $Y^{\prime }$ is 1×p. Then, from Lemma 2, $dY=\frac{\pi^{\frac{p}{2}}}{\Gamma (\frac{p}{2})}s^{\frac{p}{2}-1}ds$. Therefore, the total integral is
$$\begin{array}{cc} 1 & =\int_{X}f_{1}\left(X\right)dX\\ \; & =C_{1}\int_{X}{\left[{\left(X-\mu \right)}^{\prime }\Sigma^{-1}\left(X-\mu \right)\right]}^{\gamma }{\left[1-b\left(a-\alpha \right){\left({\left(X-\mu \right)}^{\prime }\Sigma^{-1}\left(X-\mu \right)\right)}^{\delta }\right]}^{\frac{\eta }{a-\alpha }}dX\\ \; & =C_{1}{\left|\Sigma \right|}^{\frac{1}{2}}\int_{Y}{\left[Y^{\prime }Y\right]}^{\gamma }{\left[1-b\left(a-\alpha \right){\left(Y^{\prime }Y\right)}^{\delta }\right]}^{\frac{\eta }{a-\alpha }}dY\\ \; & =C_{1}{\left|\Sigma \right|}^{\frac{1}{2}}\frac{\pi^{\frac{p}{2}}}{\Gamma (\frac{p}{2})}\int_{s=0}^{\infty }s^{\gamma +\frac{p}{2}-1}{\left[1-b\left(a-\alpha \right)s^{\delta }\right]}^{\frac{\eta }{a-\alpha }}ds\\ \; & =C_{1}{\left|\Sigma \right|}^{\frac{1}{2}}\frac{\pi^{\frac{p}{2}}}{\Gamma (\frac{p}{2})}\delta {\left[b\left(a-\alpha \right)\right]}^{-\frac{1}{\delta }\left(\gamma +\frac{p}{2}\right)}\frac{\Gamma \left(\frac{1}{\delta }\left(\gamma +\frac{p}{2}\right)\right)\Gamma \left(1+\frac{\eta }{a-\alpha }\right)}{\Gamma (1+\frac{\eta }{a-\alpha }+\frac{1}{\delta }\left(\gamma +\frac{p}{2}\right))} \end{array}$$
for α<a. The last step is obtained by integrating out *s* by using a real type-1 beta integral. Hence, for α<a, the normalizing constant is
(16)$$C_{1}=\frac{1}{{\left|\Sigma \right|}^{\frac{1}{2}}}\frac{\Gamma (\frac{p}{2})}{\pi^{\frac{p}{2}}}{\left[b\left(a-\alpha \right)\right]}^{\frac{1}{\delta }\left(\gamma +\frac{p}{2}\right)}\delta \frac{\Gamma (1+\frac{\eta }{a-\alpha }+\frac{1}{\delta }\left(\gamma +\frac{p}{2}\right))}{\Gamma \left(\frac{1}{\delta }\left(\gamma +\frac{p}{2}\right)\right)\Gamma \left(1+\frac{\eta }{a-\alpha }\right)}\;$$
for α<a,b>0,η>0,δ>0,γ>0,Σ>O. In a similar manner, and by integrating out *s* by using a real type-2 beta integral, we have the normalizing constant $C_{2}$ for α>a as the following:(17)$$C_{2}=\frac{1}{{\left|\Sigma \right|}^{\frac{1}{2}}}\frac{\Gamma (\frac{p}{2})}{\pi^{\frac{p}{2}}}{\left[b\left(\alpha -a\right)\right]}^{\frac{1}{\delta }\left(\gamma +\frac{p}{2}\right)}\delta \frac{\Gamma (\frac{\eta }{\alpha -a})}{\Gamma \left(\frac{1}{\delta }\left(\gamma +\frac{p}{2}\right)\right)\Gamma \left(\frac{\eta }{\alpha -a}-\frac{1}{\delta }\left(\gamma +\frac{p}{2}\right)\right)}\;$$
for $\alpha >a,\frac{\eta }{\alpha -a}-\frac{1}{\delta }\left(\gamma +\frac{p}{2}\right)>0,\gamma >0,\delta >0,\eta >0,\Sigma >O$, and for α→a we have
(18)$$C_{3}=\frac{1}{{\left|\Sigma \right|}^{\frac{1}{2}}}\frac{\Gamma (\frac{p}{2})}{\pi^{\frac{p}{2}}}{\left[b\eta \right]}^{\frac{1}{\delta }\left(\gamma +\frac{p}{2}\right)}\frac{\delta }{\Gamma (\frac{1}{\delta }\left(\gamma +\frac{p}{2}\right))}\;$$
for b>0,η>0,δ>0,γ>0,Σ>O.

Observe that the model in (9) is a multivariate generalized real type-1 beta model, (10) is a multivariate generalized real type-2 beta model, and (11) is a multivariate generalized real gamma model. For δ=2,γ=2, (11) is also a real multivariate Maxwell–Boltzmann model, and for $\delta =2,\gamma =\frac{1}{2}$, (11) is a real multivariate Rayleigh model. The corresponding densities for $Y=\Sigma^{-\frac{1}{2}}\left(X-\mu \right)$ can be called the standard real multivariate Maxwell–Boltzmann and Rayleigh densities, respectively. If the Maxwell–Boltzmann and Rayleigh densities are the stable distributions in a physical system, then the unstable or chaotic neighborhoods are available from (9) and (10), and all of the situations, the stable situation, the unstable neighborhoods, and the transitional stages can be reached through the pathway parameter α. For γ=0, the model in (9) is very useful in real multivariate reliability analysis; see [12,13]. The model in (10) for γ=0 corresponds to a multivariate version of Student-t, Cauchy, multivariate F, and related distributions; see [14].

From the normalizing constants $C_{1},C_{2},C_{3}$, one can also obtain the *h*-th moment of the ellipsoid of concentration for an arbitrary *h*. That is, for α<a,
(19)$$E{\left[b\left(a-\alpha \right){\left(X-\mu \right)}^{\prime }\Sigma^{-1}\left(X-\mu \right)\right]}^{h}=\frac{\Gamma (\frac{1}{\delta }\left(\gamma +h+\frac{p}{2}\right))}{\Gamma (\frac{1}{\delta }\left(\gamma +\frac{p}{2}\right))}\frac{\Gamma (1+\frac{\eta }{a-\alpha }+\frac{1}{\delta }\left(\gamma +\frac{p}{2}\right))}{\Gamma (1+\frac{\eta }{a-\alpha }+\frac{1}{\delta }\left(\gamma +h+\frac{p}{2}\right))}\;$$
for $ℜ(\gamma +h+\frac{p}{2})>0$. The density coming from (iv) is an H-function. For the theory and applications of the H-function, see [15]. Then, ${\left[b\left(a-\alpha \right){\left(X-\mu \right)}^{\prime }\Sigma^{-1}\left(X-\mu \right)\right]}^{\frac{1}{\delta }}$ is distributed as a real scalar type-1 beta random variable with the parameters $\left(\gamma +\frac{p}{2},1+\frac{\eta }{a-\alpha }\right)$ for α<a. Similarly, $b\left(\alpha -a\right){\left(X-\mu \right)}^{\prime }\Sigma^{-1}\left(X-\mu \right)$ has an H-function distribution, whereas ${\left[b\left(\alpha -a\right){\left(X-\mu \right)}^{\prime }\Sigma^{-1}\left(X-\mu \right)\right]}^{\frac{1}{\delta }}$ is a real scalar type-2 beta variable with the parameters $\left(\gamma +\frac{p}{2},\frac{\eta }{\alpha -a}-\left(\gamma +\frac{p}{2}\right)\right)$ for α>a, and ${\left[b\eta {\left(X-\mu \right)}^{\prime }\Sigma^{-1}\left(X-\mu \right)\right]}^{\frac{1}{\delta }}$ is a real scalar gamma random variable with the parameters $\left(\gamma +\frac{p}{2},1\right)$.

**Theorem** **1.**
*For the $f_{1}\left(X\right),f_{2}\left(X\right),f_{3}\left(X\right)$ defined in (9)–(11), respectively, $[b\left(a-\alpha \right){\left(X-\mu \right)}^{\prime }\Sigma^{-1}$${\left(X-\mu \right)]}^{\frac{1}{\delta }}$ is a real scalar type-1 beta random variable with the parameters $\left(\gamma +\frac{p}{2},1+\frac{\eta }{a-\alpha }\right)$ for α<a; ${\left[b\left(\alpha -a\right){\left(X-\mu \right)}^{\prime }\Sigma^{-1}\left(X-\mu \right)\right]}^{\frac{1}{\delta }}$ is a real scalar type-2 beta random variable with the parameters $\left(\gamma +\frac{p}{2},\frac{\eta }{\alpha -a}-\left(\gamma +\frac{p}{2}\right)\right)$ for α>a and $\frac{\eta }{\alpha -a}-\left(\gamma +\frac{p}{2}\right)>0$; ${\left[b\eta {\left(X-\mu \right)}^{\prime }\Sigma^{-1}\left(X-\mu \right)\right]}^{\frac{1}{\delta }}$ is a real scalar gamma random variable with the parameters $\left(\gamma +\frac{p}{2},1\right)$.*


**Note** **1.**
*We can relax the condition δ>0. Note that the models in (10) and (11) are also valid for δ<0, and by defining the support appropriately, we can relax the condition δ>0 in (9) as well.*


**Note** **2.**
*Consider a function $g({\left(X-\mu \right)}^{\prime }\Sigma^{-1}\left(X-\mu \right))$ for $g(r^{2})\geq 0$ for some real scalar variable r and let $\int_{r}g\left(r^{2}\right)dr<\infty$. Consider the optimization of Mathai’s entropy in (4) over all possible densities f(X) and under the constraint*
$$E{\left[g\left({\left(X-\mu \right)}^{\prime }\Sigma^{-1}\left(X-\mu \right)\right)\right]}^{\frac{a-\alpha }{\eta }}=\mathrm{fixed}$$
*over all f(X), where the expectation is taken in f(X); then, we end up with an elliptically contoured distribution for f(X) when the corresponding density for $Y=\Sigma^{-\frac{1}{2}}\left(X-\mu \right)$ is a spherically symmetric distribution that is invariant under orthonormal transformations or under the rotation of the axes of coordinates.*


## 4. Real Matrix-Variate Case

Let $X=(x_{ij})$ be a real p×q,p≤q, and rank *p* matrix with distinct real scalar variables $x_{ij}$ as elements. Let A>O be a p×p constant positive definite matrix and let B>O be a q×q constant positive definite matrix. Let $u=\mathrm{tr}(A^{\frac{1}{2}}XBX^{\prime }A^{\frac{1}{2}})$. This *u* is an important quantity in statistical literature. Hence, we will impose restrictions in terms of moments of *u*. Consider the optimization of Mathai’s entropy in (4) over all densities f(X), where *X* is a p×q matrix, as defined above, subject to the constraints:$$E\left[u^{\gamma (\frac{a-\alpha }{\eta })}\right]=\mathrm{fixed}\;\mathrm{and}\;E\left[u^{\gamma (\frac{a-\alpha }{\eta })+\delta }\right]=\mathrm{fixed}$$
over all possible densities f(X). Then, proceeding as in Section 3, we end up with the following densities, where we use the same notations of $f_{i}\left(X\right),C_{i},i=1,2,3$ in order to avoid having too many symbols: For α<a,
(20)$$f_{1}\left(X\right)=C_{1}{\left[\mathrm{tr}\left(A^{\frac{1}{2}}XBX^{\prime }A^{\frac{1}{2}}\right)\right]}^{\gamma }{\left[1-b\left(a-\alpha \right){\left(\mathrm{tr}\left(A^{\frac{1}{2}}XBX^{\prime }A^{\frac{1}{2}}\right)\right)}^{\delta }\right]}^{\frac{\eta }{a-\alpha }};\;$$
For α>a,
(21)$$f_{2}\left(X\right)=C_{2}{\left[\mathrm{tr}\left(A^{\frac{1}{2}}XBX^{\prime }A^{\frac{1}{2}}\right)\right]}^{\gamma }{\left[1+b\left(\alpha -a\right){\left(\mathrm{tr}\left(A^{\frac{1}{2}}XBX^{\prime }A^{\frac{1}{2}}\right)\right)}^{\delta }\right]}^{-\frac{\eta }{\alpha -a}}\;$$
and for α→a,
(22)$$f_{3}\left(X\right)=C_{3}{\left[\mathrm{tr}\left(A^{\frac{1}{2}}XBX^{\prime }A^{\frac{1}{2}}\right)\right]}^{\gamma }e^{-b\eta {\left(\mathrm{tr}{\left(A^{\frac{1}{2}}XBX^{\prime }A^{\frac{1}{2}}\right)}^{\delta }\right)}^{\delta }}.\;$$
For evaluating the normalizing constants, we use the following transformations: $Y=A^{\frac{1}{2}}XB^{\frac{1}{2}}\Rightarrow dY={\left|A\right|}^{-\frac{q}{2}}{\left|B\right|}^{-\frac{p}{2}}dX$, $s=\mathrm{tr}(YY^{\prime })=$ the sum of squares of all the pq elements in *Y*, and, hence, $\mathrm{tr}\left(YY^{\prime }\right)=ZZ^{\prime }$, where *Z* is a 1×pq vector. Then, from Lemma 2, $s=ZZ^{\prime }\Rightarrow dY=dZ=\frac{\pi^{\frac{pq}{2}}}{\Gamma (\frac{pq}{2})}s^{\frac{pq}{2}-1}ds.$ Then, for α<a, we evaluate the *s*-integral by using a real scalar type-1 beta integral; for α>a, we evaluate the *s*-integral by using a real scalar type-2 beta integral; for α→a, the *s*-integral is evaluated by using a real scalar gamma integral. Then, the normalizing constants are the following:
(23)$$C_{1}={\left|A\right|}^{\frac{q}{2}}{\left|B\right|}^{\frac{p}{2}}\frac{\Gamma (\frac{pq}{2})}{\pi^{\frac{pq}{2}}}\delta {\left[b\left(a-\alpha \right)\right]}^{\frac{1}{\delta }\left(\gamma +\frac{pq}{2}\right)}\frac{\Gamma (1+\frac{\eta }{a-\alpha }+\frac{1}{\delta }\left(\gamma +\frac{pq}{2}\right))}{\Gamma \left(\frac{1}{\delta }\left(\gamma +\frac{pq}{2}\right)\right)\Gamma \left(1+\frac{\eta }{a-\alpha }\right)}$$
(24)$$C_{2}={\left|A\right|}^{\frac{q}{2}}{\left|B\right|}^{\frac{p}{2}}\frac{\Gamma (\frac{pq}{2})}{\pi^{\frac{pq}{2}}}\delta {\left[b\left(\alpha -a\right)\right]}^{\frac{1}{\delta }\left(\gamma +\frac{pq}{2}\right)}\frac{\Gamma (\frac{\eta }{\alpha -a})}{\Gamma \left(\frac{1}{\delta }\left(\gamma +\frac{pq}{2}\right)\right)\Gamma \left(\frac{\eta }{\alpha -a}-\frac{1}{\delta }\left(\gamma +\frac{pq}{2}\right)\right)}$$
(25)$$C_{3}={\left|A\right|}^{\frac{q}{2}}{\left|B\right|}^{\frac{p}{2}}\frac{\Gamma (\frac{pq}{2})}{\pi^{\frac{pq}{2}}}\delta {\left[b\eta \right]}^{\frac{1}{\delta }\left(\gamma +\frac{pq}{2}\right)}\frac{1}{\Gamma (\frac{1}{\delta }\left(\gamma +\frac{pq}{2}\right))}$$
where, in (23), the conditions are α<a,A>O,B>O,b>0,η>0,δ>0,γ>0; in (24), the conditions are $\alpha >a,A>O,B>O,b>0,\eta >0,\delta >0,\gamma >0,\frac{\eta }{\alpha -a}-\frac{1}{\delta }\left(\gamma +\frac{pq}{2}\right)>0$; in (25), the conditions are A>O,B>O,b>0,η>0,γ>0,δ>0.

Observe that (21) and (22) are available from (20). Similarly, (20) and (22) are available from (21). In other words, all densities in (20)–(22) are available through the pathway parameter α. Note that (22) for δ=1,γ=1 can be taken as a multivariate version of Maxwell–Boltzmann density coming from a rectangular matrix-variate real random variable. Similarly, for $\delta =1,\gamma =\frac{1}{2}$, one can take (22) as a version of the multivariate real Rayleigh density coming from a rectangular matrix-variate real random variable. For γ=0,δ=1, (20) is a real rectangular matrix-variate Gaussian density. One can consider (20) as a generalized real multivariate type-1 beta density, (21) as a generalized real multivariate type-2 beta density, and (22) as the corresponding gamma density. For γ=0, the model in (20) is a suitable model for reliability analysis for a real multivariate situation. As observed in Section 3, one can see that $\mathrm{tr}(A^{\frac{1}{2}}XBX^{\prime }A^{\frac{1}{2}})$ has an H-function distribution for α<a,α>a,α→a. In addition, for α<a, ${\left[b\left(\alpha -a\right)\mathrm{tr}\left(A^{\frac{1}{2}}XBX^{\prime }A^{\frac{1}{2}}\right)\right]}^{\frac{1}{\delta }}$ is a real scalar type-1 beta distributed with the parameters $\left(\gamma +\frac{pq}{2},1+\frac{\eta }{a-\alpha }\right)$; for α>a, ${\left[b\left(\alpha -a\right)\mathrm{tr}\left(A^{\frac{1}{2}}XBX^{\prime }A^{\frac{1}{2}}\right)\right]}^{\frac{1}{\delta }}$ is a real scalar type-2 beta distributed with the parameters $\left(\gamma +\frac{pq}{2},\frac{\eta }{\alpha -a}-\left(\gamma +\frac{pq}{2}\right)\right)$; for α→a, ${\left[b\eta \mathrm{tr}\left(A^{\frac{1}{2}}XBX^{\prime }A^{\frac{1}{2}}\right)\right]}^{\frac{1}{\delta }}$ is a real scalar gamma distributed with the parameters $\left(\gamma +\frac{pq}{2},1\right)$. 

**Note** **3.**
*If a location parameter p×q matrix M is to be introduced, then replace X with X−M everywhere. If q≤p and if X is of rank q, then one can consider $v=\mathrm{tr}(B^{\frac{1}{2}}X^{\prime }AXB^{\frac{1}{2}})$. Then, parallel results hold for all of the results in Section 4 by interchanging A with B and p with q.*


## 5. Constraints in Terms of Determinants

Let $X=(x_{ij})$ be a p×q,p≤q, and rank *p* matrix with distinct elements $x_{ij}$. Let A>O be p×p and B>O be q×q constant positive definite matrices. Consider the optimization of Mathai’s entropy (4) under the constraint
$$E[|I-b\left(a-\alpha \right)A^{\frac{1}{2}}XBX^{\prime }A^{\frac{1}{2}}|]=\mathrm{fixed}$$
over all real p×q,p≤q, and rank *p* matrix-variate densities f(X). Then, following the same procedure as in the above cases, we end up with the density
(26)$$f_{1}\left(X\right)=C_{1}{\left|I-b\left(a-\alpha \right)A^{\frac{1}{2}}XBX^{\prime }A^{\frac{1}{2}}\right|}^{\frac{\eta }{a-\alpha }}\;$$
for $\alpha <a,b>0,\eta >0,A>O,B>O,I-b\left(a-\alpha \right)A^{\frac{1}{2}}XBX^{\prime }A^{\frac{1}{2}}>O$, and *a* is a fixed scalar constant. In order to avoid having too many symbols, we will use the same notations of $f_{i}\left(X\right),C_{i},i=1,2,3$ in this section. For α>a, the model in (26) changes into
(27)$$f_{2}\left(X\right)=C_{2}{\left|I+b\left(\alpha -a\right)A^{\frac{1}{2}}XBX^{\prime }A^{\frac{1}{2}}\right|}^{-\frac{\eta }{\alpha -a}}\;$$
for α>a,b>0,η>0,A>O,B>O. When α→a, both $f_{1}\left(X\right)$ and $f_{2}\left(X\right)$ go to
(28)$$f_{3}\left(X\right)=C_{3}e^{-b\eta \mathrm{tr}(A^{\frac{1}{2}}XBX^{\prime }A^{\frac{1}{2}})}\;$$
for b>0,η>0,A>O,B>O. The transition of (26) and (27) to (28) can be seen from the following properties. Let $\lambda_{1},\ldots ,\lambda_{p}$ be the eigenvalues of $A^{\frac{1}{2}}XBX^{\prime }A^{\frac{1}{2}}$. Then,
$$|I-b\left(a-\alpha \right)A^{\frac{1}{2}}XBX^{\prime }A^{\frac{1}{2}}{|}^{\frac{\eta }{a-\alpha }}=\prod_{j=1}^{p}{\left[1-b\left(a-\alpha \right)\lambda_{j}\right]}^{\frac{\eta }{a-\alpha }}.$$
However, from the definition of the mathematical constant *e*, we have $\lim_{\alpha \to a_{-}}{\left(1-b\left(a-\alpha \right)\lambda_{j}\right)}^{\frac{\eta }{a-\alpha }}=e^{-b\eta \lambda_{j}}$. In a similar manner, $\lim_{\alpha \to a_{+}}{\left(1+b\left(\alpha -a\right)\lambda_{j}\right)}^{-\frac{\eta }{\alpha -a}}=e^{-b\eta \lambda_{j}}$. Then, the product gives the sum of the eigenvalues or the trace in the exponent and, hence, the result. The normalizing constants $C_{i},i=1,2,3$ can be evaluated by using the following transformations: $Y=A^{\frac{1}{2}}XB^{\frac{1}{2}}\Rightarrow dY={\left|A\right|}^{\frac{q}{2}}{\left|B\right|}^{\frac{p}{2}}dX$ by using Lemma 1 or $A=YY^{\prime }\Rightarrow dY=\frac{\pi^{\frac{pq}{2}}}{\Gamma_{p}\left(\frac{q}{2}\right)}{\left|S\right|}^{\frac{q}{2}-\frac{p+1}{2}}dS$ by using Lemma 2. Then, evaluating the *S*-integral by using a real matrix-variate type-1 beta integral for α<a, by using a real matrix-variate type-2 beta integral for α>a, or by using a real matrix-variate gamma integral for α→a, we obtain the results, where, for example, $\Gamma_{p}\left(\alpha \right)$ is the real matrix-variate gamma defined earlier in (14).

### 5.1. Modification of the Constraint in Terms of a Determinant

Let us consider the matrices X,A,B as in Section 5. Consider the optimization of (4) under the following constraint for α<a:$$E[|A^{\frac{1}{2}}XBX^{\prime }A^{\frac{1}{2}}{|}^{\gamma \frac{\left(a-\alpha \right)}{\eta }}|I-b\left(a-\alpha \right)A^{\frac{1}{2}}XBX^{\prime }A^{\frac{1}{2}}|]=\mathrm{fixed}$$
over all possible densities f(X). Then, proceeding as in the previous cases, we end up with the following densities:(29)$$f_{1}\left(X\right)=C_{1}|A^{\frac{1}{2}}XBX^{\prime }A^{\frac{1}{2}}{|}^{\gamma }{\left|I-b\left(a-\alpha \right)A^{\frac{1}{2}}XBX^{\prime }A^{\frac{1}{2}}\right|}^{\frac{\eta }{a-\alpha }}\;$$
for $\alpha <a,b>0,\gamma >0,\eta >o,A>O,B>O,I-b\left(a-\alpha \right)A^{\frac{1}{2}}XBX^{\prime }A^{\frac{1}{2}}>O$. For α>a, we have
(30)$$f_{2}\left(X\right)=C_{2}|A^{\frac{1}{2}}XBX^{\prime }A^{\frac{1}{2}}{|}^{\gamma }{\left|I+b\left(\alpha -a\right)A^{\frac{1}{2}}XBX^{\prime }A^{\frac{1}{2}}\right|}^{-\frac{\eta }{\alpha -a}}\;$$
for α>a,b>0,η>0,γ>0,A>O,B>O, and for α→a, we have
(31)$$f_{2}\left(X\right)=C_{3}{\left|A^{\frac{1}{2}}XBX^{\prime }A^{\frac{1}{2}}\right|}^{\gamma }e^{-b\eta \mathrm{tr}(A^{\frac{1}{2}}XBX^{\prime }A^{\frac{1}{2}})}\;$$
for b>0,η>0,A>O,B>O. Observe that, as in the previous cases, all three models are available through the pathway parameter α from either $f_{1}\left(X\right)$ or $f_{2}\left(X\right)$. If $f_{3}\left(X\right)$ is the stable situation in a physical system, then the unstable neighborhoods are given by $f_{1}\left(X\right)$ and $f_{2}\left(X\right)$; these stable and unstable stages and the transitional stages can be reached through α. For γ=1, the model in (31) can be taken as the real rectangular matrix-variate Maxwell–Boltzmann density, and for $\gamma =\frac{1}{2}$, it is the real rectangular matrix-variate Rayleigh density. The corresponding densities of $Y=A^{\frac{1}{2}}XB^{\frac{1}{2}}$ can be taken as the standard matrix-variate Maxwell–Boltzmann and Rayleigh densities. The corresponding densities for $S=YY^{\prime }$ can be taken as the isotropic or spherically symmetric matrix-variate Maxwell–Boltzmann and Rayleigh densities. The normalizing constants can be evaluated by using the transformations in Section 5 and then evaluating the *S*-integral by using real matrix-variate type-1 beta, type-2 beta, and gamma integrals. The final expressions will be the following:
(32)$$C_{1}={\left|A\right|}^{\frac{q}{2}}{\left|B\right|}^{\frac{p}{2}}{\left[b\left(a-\alpha \right)\right]}^{p\gamma +\frac{pq}{2}}\frac{\Gamma_{p}\left(\frac{q}{2}\right)}{\pi^{\frac{pq}{2}}}\frac{\Gamma_{p}\left(\gamma +\frac{q}{2}+\frac{p+1}{2}+\frac{\eta }{a-\alpha }\right)}{\Gamma_{p}\left(\gamma +\frac{q}{2}\right)\Gamma_{p}\left(\frac{p+1}{2}+\frac{\eta }{a-\alpha }\right)}$$
(33)$$C_{2}={\left|A\right|}^{\frac{q}{2}}{\left|B\right|}^{\frac{p}{2}}{\left[b\left(\alpha -a\right)\right]}^{p\gamma +\frac{pq}{2}}\frac{\Gamma_{p}\left(\frac{q}{2}\right)}{\pi^{\frac{pq}{2}}}\frac{\Gamma_{p}\left(\frac{\eta }{\alpha -a}\right)}{\Gamma_{p}\left(\gamma +\frac{q}{2}\right)\Gamma_{p}\left(\frac{\eta }{\alpha -a}-\gamma -\frac{q}{2}\right)}$$
(34)$$C_{3}={\left|A\right|}^{\frac{q}{2}}{\left|B\right|}^{\frac{p}{2}}{\left[b\eta \right]}^{p\gamma +\frac{pq}{2}}\frac{\Gamma_{p}\left(\frac{q}{2}\right)}{\pi^{\frac{pq}{2}}}\frac{1}{\Gamma_{p}\left(\gamma +\frac{q}{2}\right)}$$
where in (32) α<a, in (33) α>a and $\frac{\eta }{\alpha -a}-\left(\gamma +\frac{q}{2}\right)>0$, and in (32)–(34) A>O,B>O,γ>0,η>0,γ>0,b>0.

**Note** **4.**
*If a location parameter is needed, then replace X with X−M, where M is a p×q constant matrix everywhere in Section 5 and Section 5.1.*


### 5.2. Arbitrary Moments

Let $u=|A^{\frac{1}{2}}XBX^{\prime }A^{\frac{1}{2}}|$, and if the *h*-th moment of this determinant *u* for an arbitrary *h* is needed, then this moment can be written down by looking at the normalizing constants in (32)–(34). For α<a, the *h*-th moment is the following: $$\begin{array}{cc} E[u^{h}] & ={\left[b\left(a-\alpha \right)\right]}^{-p(\gamma +\frac{q}{2})}\frac{\Gamma_{p}\left(\gamma +\frac{q}{2}+h\right)}{\Gamma_{p}\left(\gamma +\frac{q}{2}\right)}\frac{\Gamma_{p}\left(\frac{p+1}{2}+\frac{\eta }{a-\alpha }+\gamma +\frac{q}{2}\right)}{\Gamma_{p}\left(\frac{p+1}{2}+\frac{\eta }{a-\alpha }+\gamma +\frac{q}{2}+h\right)}\\ \; & ={\left[b\left(a-\alpha \right)\right]}^{-p(\gamma +\frac{q}{2})}\prod_{j=1}^{p}\frac{\Gamma (\gamma +\frac{q}{2}+h-\frac{j-1}{2})}{\Gamma (\gamma +\frac{q}{2}-\frac{j-1}{2})}\frac{\Gamma (\frac{p+1}{2}+\frac{\eta }{a-\alpha }+\gamma +\frac{q}{2}-\frac{j-1}{2})}{\Gamma (\frac{p+1}{2}+\frac{\eta }{a-\alpha }+\gamma +\frac{q}{2}-\frac{j-1}{2}+h)}\\ E{\left[b\left(a-\alpha \right)u\right]}^{h} & =E\left[u_{1}^{h}\right]E\left[u_{2}^{h}\right]...E\left[u_{p}^{h}\right] \end{array}$$
for $ℜ\left(h+\gamma +\frac{q}{2}\right)>\frac{p-1}{2}$, where $u_{1},\ldots ,u_{p}$ are independently distributed real scalar type-1 beta random variables, with $u_{j}$ having the parameters $(\gamma +\frac{q}{2}-\frac{j-1}{2},\frac{p+1}{2}+\frac{\eta }{a-\alpha }),j=1,\ldots ,p$, so that we have the following structural representation for α<a:(35)$$|b\left(a-\alpha \right)A^{\frac{1}{2}}XBX^{\prime }A^{\frac{1}{2}}|=u_{1}\ldots u_{p}.\;$$

Both sides have the same distribution. Similarly, for α>a, we have the following:
(36)$$\begin{array}{cc} E{\left[b\left(\alpha -a\right)u\right]}^{h} & =\prod_{j=1}^{p}\frac{\Gamma (\gamma +\frac{q}{2}-\frac{j-1}{2}+h)}{\Gamma (\gamma +\frac{q}{2}-\frac{j-1}{2})}\frac{\Gamma (\frac{\eta }{\alpha -a}-\left(\gamma +\frac{q}{2}\right)-\frac{j-1}{2})}{\Gamma (\frac{\eta }{\alpha -a}-\left(\gamma +\frac{q}{2}\right)-\frac{j-1}{2}-h)}\\ \; & =E\left[v_{1}^{h}\right]E\left[v_{2}^{h}\right]\ldots E\left[v_{p}^{h}\right]\Rightarrow \\ b(\alpha -a)u & =v_{1}v_{2}\ldots v_{p} \end{array}$$
for $ℜ\left(h+\gamma +\frac{q}{2}\right)>\frac{p-1}{2},\frac{\eta }{\alpha -a}-\left(\gamma +\frac{q}{2}\right)-\frac{j-1}{2}>0,j=1,\ldots ,p$, where $v_{1},\ldots ,v_{p}$ are independently distributed real scalar type-2 beta variables, with $v_{j}$ having the parameters $(\gamma +\frac{q}{2}-\frac{j-1}{2},\frac{\eta }{\alpha -a}-\left(\gamma +\frac{q}{2}\right)-\frac{j-1}{2}),j=1,\ldots ,p$. For α→a, we have the following from (34):
(37)$$\begin{array}{cc} E{\left[b\eta \;u\right]}^{h} & =\frac{\Gamma_{p}\left(\gamma +\frac{q}{2}+h\right)}{\Gamma_{p}\left(\gamma +\frac{q}{2}\right)}=\prod_{j=1}^{p}\frac{\Gamma (\gamma +\frac{q}{2}-\frac{j-1}{2}+h)}{\Gamma (\gamma +\frac{q}{2}-\frac{j-1}{2})}\\ \; & =E\left[w_{1}^{h}\right]E\left[w_{2}^{h}\right]\ldots E\left[w_{p}^{h}\right]\Rightarrow \\ b\eta \;u & =w_{1}\ldots w_{p} \end{array}$$
for $ℜ\left(h+\gamma +\frac{q}{2}\right)>\frac{p-1}{2}$, where $w_{1},\ldots ,w_{p}$ are independently distributed real scalar gamma variables, with $w_{j}$ having the parameters $(\gamma +\frac{q}{2}-\frac{j-1}{2},1),j=1,\ldots ,p$. 

**Note** **5.**
*Note that $u=|A^{\frac{1}{2}}XBX^{\prime }A^{\frac{1}{2}}|=|YY^{\prime }|$. Let the rows of Y be $Y_{1},\ldots ,Y_{p}$, where $Y_{j}$ is a 1×q real vector. Then, $Y_{j}$ can be considered to be a point in a q-dimensional Euclidean space. We have p≤q of such points. These points (vectors) are linearly independent because we have assumed that the matrix is of rank p. Taking the points in the order $Y_{1},\ldots ,Y_{p}$, these points (vectors) create a convex hull, and in this hull, a parallelotope is determined; the volume content of this parallelotope is the determinant $|YY^{\prime }{|}^{\frac{1}{2}}$. Hence, the distribution of this determinant, as well as the moments, is important in stochastic geometry or in geometrical probabilities and other related areas of image processing, pattern recognition, etc. The scaling constants b(a−α) in (35), b(α−a) in (36), and bη in (37) can be taken as unities for convenience. Then, the points $Y_{1},\ldots ,Y_{p}$ are type-1 beta distributed in (35), type-2 beta distributed in (36), and gamma distributed in (37). In general, $Y_{1},\ldots ,Y_{p}$ have pathway distributions, or these are pathway-distributed random points in q-space.*


**Note** **6.**
*If q≤p and the matrix X is of rank q, then we may consider $B^{\frac{1}{2}}X^{\prime }AXB^{\frac{1}{2}}$. Then, results corresponding to the results in Section 5, Section 5.1 and Section 5.2 are available by interchanging A with B and p with q. Hence, a separate discussion is not needed in this case. Observe also that $\mathrm{tr}\left(A^{\frac{1}{2}}XBX^{\prime }A^{\frac{1}{2}}\right)=\mathrm{tr}\left(B^{\frac{1}{2}}X^{\prime }AXB^{\frac{1}{2}}\right)$, where one is a p×p matrix and the other is a q×q matrix.*


## 6. Complex Case

For a matrix *A*, its transpose will be written as $A^{\prime }$ and its complex conjugate transpose as $A^{*}$. If $A=A^{*}$, then *A* is called Hermitian. Any complex matrix *A* can be written as $A=A_{1}+iA_{2},i=\sqrt{\left(-1\right)},A_{1},A_{2}$ real. When *A* is Hermitian, then $A_{1}=A_{1}^{\prime },A_{2}=-A_{2}^{\prime }$, that is, $A_{1}$ is real symmetric and $A_{2}$ is real skew symmetric. If *A* is p×p Hermitian positive definite, then $A=A^{*}>O$ (Hermitian positive definite). The determinant of *A* is written as |A|, as well as det(A), and the absolute value of the determinant will be written as $\left|\det\left(A\right)\right|=\sqrt{\left[\det\left(A\right)\right]\left[\det\left(A^{*}\right)\right]}=\sqrt{\left[\det\left(AA^{*}\right)\right]}$. Variables in the complex domain will be written with a tilde, such as $\tilde{X}$. In order to optimize Mathai’s entropy (4) over the density $f(\tilde{X})$ in the complex domain, we need some results on Jacobians. These will be given here as lemmas without proofs. For the proofs and for other related results in the complex domain, see [11].

**Lemma** **3.**
*Let $\tilde{X}=\left({\tilde{x}}_{ij}\right)$ be p×q with distinct complex scalar elements ${\tilde{x}}_{ij}$. Let A and B be p×p or q×q nonsingular constant matrices, respectively—real or complex. Then,*
(38)$$\tilde{Y}=A\tilde{X}B,|A|\neq 0,|B|\neq 0\Rightarrow d\tilde{Y}=|\det\left(AA^{*}\right){|}^{q}{\left|\det\left(BB^{*}\right)\right|}^{p}d\tilde{X}.\;$$


**Lemma** **4.**
*Let $\tilde{X}=\left({\tilde{x}}_{ij}\right)$ be a p×q,p≤q, and rank p matrix with distinct complex elements ${\tilde{x}}_{ij}$. Let $\tilde{S}=\tilde{X}{\tilde{X}}^{*}$, which is p×p Hermitian positive definite. Then, after integrating over the Stiefel manifold,*
(39)$$d\tilde{X}=\frac{\pi^{pq}}{{\tilde{\Gamma }}_{p}\left(q\right)}{\left|\det\left(\tilde{S}\right)\right|}^{q-p}d\tilde{S}\;$$
*where ${\tilde{\Gamma }}_{p}\left(\alpha \right)$ is a complex matrix-variate gamma given by*
(40)$${\tilde{\Gamma }}_{p}\left(\alpha \right)=\pi^{\frac{p(p-1)}{2}}\Gamma \left(\alpha \right)\Gamma \left(\alpha -1\right)...\Gamma \left(\alpha -p+1\right),ℜ\left(\alpha \right)>p-1$$
(41)$$\;=\int_{\tilde{S}>O}{\left|\det\left(\tilde{S}\right)\right|}^{\alpha -p}e^{-\mathrm{tr}(\tilde{S})}d\tilde{S},ℜ\left(\alpha \right)>p-1.$$


### Optimization in the Complex Domain

As a first problem, let $\tilde{X}$ be a p×1 vector variable in the complex domain with distinct scalar complex elements. Let Σ>O be a p×p Hermitian positive definite constant matrix. Consider the Hermitian form $u={\left(\tilde{X}-\mu \right)}^{*}\Sigma^{-1}\left(\tilde{X}-\mu \right)$, where μ is a p×1 constant vector. This can be taken as the ellipsoid of concentration in 2p-dimensional Euclidean space or as the ellipsoid of concentration in a *p*-dimensional complex domain. This ellipsoid is an important quantity in statistical analysis, as well as in various other situations. When $\tilde{X}$ is a vector random variable in the complex domain with the mean value $E[\tilde{X}]=\mu$ and with the covariance matrix $\Sigma =\mathrm{Cov}\left(\tilde{X}\right)=E\left[\left(\tilde{X}-\mu \right){\left(\tilde{X}-\mu \right)}^{*}\right]$, then *u* is the generalized distance of $\tilde{X}$ from the point of the location of its expected value μ. Hence, we will optimize Mathai’s entropy in (4) under moment-like constraints on *u*. Consider the following constraints: For α<a,
$$E\left[u^{\gamma (\frac{a-\alpha }{\eta })}\right]=\mathrm{fixed}\;\mathrm{and}\;E\left[u^{\gamma (\frac{a-\alpha }{\eta })+\delta }\right]=\mathrm{fixed}$$
over all possible densities $f(\tilde{X})$, where a,α,η>0,δ>0,γ>0 are all real scalar constants, *a* is a fixed location, and α is a real parameter. For α<a, proceeding as in the real case, we will end up with the following density:(42)$$f_{1}\left(\tilde{X}\right)=c_{1}{\left[{\left(\tilde{X}-\mu \right)}^{*}\Sigma^{-1}\left(\tilde{X}-\mu \right)\right]}^{\gamma }{\left[1-b\left(a-\alpha \right){\left({\left(\tilde{X}-\mu \right)}^{*}\Sigma^{-1}\left(\tilde{X}-\mu \right)\right)}^{\delta }\right]}^{\frac{\eta }{a-\alpha }}\;$$
for $1-b\left(a-\alpha \right){\left(\tilde{X}-\mu \right)}^{*}\Sigma^{-1}\left(\tilde{X}-\mu \right)>0$, where $c_{1}$ is the normalizing constant. In order to avoid having too many symbols, we will use the same notations as in the real case, with variables written with a tilde and constants without a tilde. For α>a, we will have the following density:
(43)$$f_{2}\left(\tilde{X}\right)=c_{2}{\left[{\left(\tilde{X}-\mu \right)}^{*}\Sigma^{-1}\left(\tilde{X}-\mu \right)\right]}^{\gamma }{\left[1+b\left(\alpha -a\right){\left({\left(\tilde{X}-\mu \right)}^{*}\Sigma^{-1}\left(\tilde{X}-\mu \right)\right)}^{\delta }\right]}^{-\frac{\eta }{\alpha -a}}\;$$
for α>a,b>0,η>0,δ>0,γ>0. When α→a, both $f_{1}\left(\tilde{X}\right)$ and $f_{2}\left(\tilde{X}\right)$ go to
(44)$$f_{3}\left(\tilde{X}\right)=c_{3}{\left[{\left(\tilde{X}-\mu \right)}^{*}\Sigma^{-1}\left(\tilde{X}-\mu \right)\right]}^{\gamma }e^{-b\eta {\left(\tilde{X}-\mu \right)}^{*}\Sigma^{-1}\left(\tilde{X}-\mu \right)}\;$$
for b>0,η>0. For evaluating the normalizing constants $c_{1},c_{2},c_{3}$, we will use the following transformations: $\tilde{Y}=\Sigma^{-\frac{1}{2}}\left(\tilde{X}-\mu \right)\Rightarrow d\tilde{Y}={\left|\det\left(\Sigma \right)\right|}^{-1}d\tilde{X}$ by using Lemma 3, where $\Sigma^{-\frac{1}{2}}$ is the Hermitian positive definite square root of the Hermitian positive definite $\Sigma^{-1}$; $s=Y^{*}Y\Rightarrow dY=\frac{\pi^{q}}{\tilde{\Gamma }\left(q\right)}s^{q-1}ds$ by using Lemma 4, where ${\tilde{Y}}^{*}$ is 1×p, and *s* is 1×1. Then, we evaluate the *s*-integral by using a real scalar type-1 beta integral for α<a, real scalar type-2 beta integral for α>a, and real scalar gamma integral for α→a. Then, we have the following results:
(45)$$c_{1}=\frac{\Gamma (p)}{\left|\det\left(\Sigma \right)\right|\pi^{p}}\delta {\left[b\left(a-\alpha \right)\right]}^{\frac{\gamma +p}{\delta }}\frac{\Gamma (1+\frac{\eta }{a-\alpha }+\frac{\gamma +p}{\delta })}{\Gamma \left(\frac{\gamma +p}{\delta }\right)\Gamma \left(1+\frac{\eta }{a-\alpha }\right)},\alpha <a$$
(46)$$c_{2}=\frac{\Gamma (p)}{\left|\det\left(\Sigma \right)\right|\pi^{p}}\delta {\left[b\left(\alpha -a\right)\right]}^{\frac{\gamma +p}{\delta }}\frac{\Gamma (\frac{\eta }{\alpha -a})}{\Gamma \left(\frac{\gamma +p}{\delta }\right)\Gamma \left(\frac{\eta }{\alpha -a}-\left(\frac{\gamma +p}{\delta }\right)\right)},\alpha >a$$
(47)$$c_{3}=\frac{\Gamma (p)}{\left|\det\left(\Sigma \right)\right|\pi^{p}}\delta {\left[b\eta \right]}^{\frac{\gamma +p}{\delta }}\frac{1}{\Gamma (\frac{\gamma +p}{\delta })}$$
for b>0,η>0,γ>0,δ>0, and in addition, in (46), $\frac{\eta }{\alpha -a}-\left(\frac{\gamma +p}{\delta }\right)>0$. Observe that through the pathway parameter α, one can reach all three densities $f_{j}\left(\tilde{X}\right),j=1,2,3$, and hence, $f_{1}\left(\tilde{X}\right)$ or $f_{2}\left(\tilde{X}\right)$ is the pathway model in the complex domain for the p×1 vector random variable $\tilde{X}$. In model-building situations, if $f_{3}\left(\tilde{X}\right)$ is the stable model, then the unstable neighborhoods are given by $f_{1}\left(\tilde{X}\right)$ and $f_{2}\left(\tilde{X}\right)$, and the transitional stages are also reached through α.

For γ=1,δ=1, one can consider $f_{3}\left(\tilde{X}\right)$ in (44) as a multivariate Maxwell–Boltzmann density in the complex domain. For $\gamma =\frac{1}{2},\delta =1$, one can take (44) as a multivariate Rayleigh density in the complex domain. For p=1, we have the scalar variable Maxwell–Boltzmann and Rayleigh densities in the complex domain from (44). The corresponding real cases may be seen from [12,13]. Observe that (43) and (44) also hold for δ<0, but for δ<0, the support must be redefined in (42). Hence, a form of multivariate Maxwell–Boltzmann and Rayleigh densities can be defined for δ<0 as well. In the complex domain, these densities are defined over the whole complex space. In the complex scalar case, if one has to confine to the sector $ℜ(\tilde{x}-\mu )>0$, then we multiply the corresponding (44) by $\frac{1}{2}$ for p=1 so that one can consider, for example, a time variable that is real positive for the real part and a phase variable for the complex part. Note that in the Rayleigh case for $p=1,\gamma =\frac{1}{2}$, we have ${\left[{\left(\tilde{x}-\mu \right)}^{*}\left(\tilde{x}-\mu \right)\right]}^{\frac{1}{2}}=\left|\left(\tilde{x}-\mu \right)\right|=$, which is the absolute value. For γ=0, (42) gives a very good model for multivariate reliability analysis in the complex domain. Reliability analysis in the complex domain does not seem to have been discussed in the literature.

## 7. Optimization with a Trace Constraint

Let $\tilde{X}$ be a p×q,p≤q, and rank *p* matrix with distinct complex scalar variables as elements. Let A>O and B>O be p×p and q×q Hermitian positive definite constant matrices, respectively. Let $u=\mathrm{tr}(A^{\frac{1}{2}}\tilde{X}B{\tilde{X}}^{*}A^{\frac{1}{2}})$, where $A^{\frac{1}{2}}$ is a Hermitian positive definite square root of *A*. Consider the optimization of Mathai’s entropy in (4) for the density $f(\tilde{X})$, where $\tilde{X}$ is p×q, as described here, subject to the constraints
$$E{\left[\mathrm{tr}\left(A^{\frac{1}{2}}\tilde{X}B{\tilde{X}}^{*}A^{\frac{1}{2}}\right)\right]}^{\gamma (\frac{a-\alpha }{\eta })}=\mathrm{fixed}\;\mathrm{and}\;E{\left[\mathrm{tr}\left(A^{\frac{1}{2}}\tilde{X}B{\tilde{X}}^{*}A^{\frac{1}{2}}\right)\right]}^{\gamma (\frac{a-\alpha }{\eta })+\delta }=\mathrm{fixed}$$
over all possible densities $f(\tilde{X})$, where $\tilde{X}$ is p×q,p≤q, and of rank *p*. Then, proceeding as in the real case, we end up with the following densities: For α<a,
(48)$$f_{1}\left(\tilde{X}\right)=C_{1}{\left[u\right]}^{\gamma }{\left[1-b\left(a-\alpha \right)u^{\delta }\right]}^{\frac{\eta }{a-\alpha }},\alpha <a$$
(49)$$f_{2}\left(\tilde{X}\right)=C_{2}{\left[u\right]}^{\gamma }{\left[1+b\left(\alpha -a\right)u^{\delta }\right]}^{-\frac{\eta }{\alpha -a}},\alpha >a$$
(50)$$f_{3}\left(\tilde{X}\right)=C_{3}{\left[u\right]}^{\gamma }e^{-b\eta u^{\delta }}$$
where $u=\mathrm{tr}(A^{\frac{1}{2}}\tilde{X}B{\tilde{X}}^{*}A^{\frac{1}{2}})$, and the normalizing constants are the following:(51)$$C_{1}={\left|\det\left(A\right)\right|}^{q}{\left|\det\left(B\right)\right|}^{p}\frac{\Gamma (pq)}{\pi^{pq}}\delta {\left[b\left(a-\alpha \right)\right]}^{\frac{\gamma +pq}{\delta }}\frac{\Gamma (\frac{\gamma +pq}{\delta }+1+\frac{\eta }{a-\alpha })}{\Gamma \left(\frac{\gamma +pq}{\delta }\right)\Gamma \left(1+\frac{\eta }{\alpha -a}\right)},\;$$
for α<a,b>0,γ>0,δ>0,A>O,B>O,η>0;
(52)$$C_{2}={\left|\det\left(A\right)\right|}^{q}{\left|\det\left(B\right)\right|}^{p}\frac{\Gamma (pq)}{\pi^{pq}}\delta {\left[b\left(\alpha -a\right)\right]}^{\frac{\gamma +pq}{\delta }}\frac{\Gamma (\frac{\eta }{\alpha -a})}{\Gamma \left(\frac{\gamma +pq}{\delta }\right)\Gamma \left(\frac{\eta }{\alpha -a}-\left(\frac{\gamma +pq}{\delta }\right)\right)},\;$$
for $\alpha >a,b>0,\eta >0,\gamma >0,\delta >0,A>O,B>O,\frac{\eta }{\alpha -a}-\left(\frac{\gamma +pq}{\delta }\right)>0$ and
(53)$$C_{3}={\left|\det\left(A\right)\right|}^{q}{\left|\det\left(B\right)\right|}^{p}\frac{\Gamma (pq)}{\pi^{pq}}\delta {\left[b\eta \right]}^{\frac{\gamma +pq}{\delta }}\frac{1}{\Gamma (\frac{\gamma +pq}{\delta })}\;$$
for A>O,B>O,b>0,η>0,γ>0,δ>0.

Note that (50) can be considered as a multivariate version of the complex Maxwell–Boltzmann and Rayleigh densities for (γ=1,δ=1) and $\left(\gamma =\frac{1}{2},\delta =1\right)$, respectively. If q≤p and $\tilde{X}$ is of rank *q*, then we can take $u=\mathrm{tr}(B^{\frac{1}{2}}{\tilde{X}}^{*}A\tilde{X}B^{\frac{1}{2}})$ and proceed, as in the p≤q case, with *p* and *q* interchanged and *A* and *B* interchanged. We obtain results parallel to the ones above for the case of p≤q.

## 8. Constraints in Terms of Determinants

Let $\tilde{X}$ be p×q,p≤q, and of rank *p* with distinct complex scalar variables as elements. Let A>O and B>O be p×p and q×q Hermitian positive definite constant matrices, respectively. Consider the optimization of (4) under the constraint
$$E[|\det\left(A^{\frac{1}{2}}\tilde{X}B{\tilde{X}}^{*}A^{\frac{1}{2}}\right){|}^{\gamma (\frac{a-\alpha }{\eta })}|\det\left(I-b\left(a-\alpha \right)A^{\frac{1}{2}}\tilde{X}B{\tilde{X}}^{*}A^{\frac{1}{2}}\right)|]=\mathrm{fixed}$$
over all possible densities $f(\tilde{X})$, where $\tilde{X}$ is p×q,p≤q, and of rank *p*. Then, proceeding as in the real case, we have the following densities:
(54)$$f_{1}\left(\tilde{X}\right)=C_{1}|\det\left(A^{\frac{1}{2}}\tilde{X}B{\tilde{X}}^{*}A^{\frac{1}{2}}\right){|}^{\gamma }{\left|\det\left(I-b\left(a-\alpha \right)A^{\frac{1}{2}}\tilde{X}B{\tilde{X}}^{*}A^{\frac{1}{2}}\right)\right|}^{\frac{\eta }{a-\alpha }},\alpha <a$$
(55)$$f_{2}\left(\tilde{X}\right)=C_{2}|\det\left(A^{\frac{1}{2}}\tilde{X}B{\tilde{X}}^{*}A^{\frac{1}{2}}\right){|}^{\gamma }{\left|\det\left(I+b\left(\alpha -a\right)A^{\frac{1}{2}}\tilde{X}B{\tilde{X}}^{*}A^{\frac{1}{2}}\right)\right|}^{-\frac{\eta }{\alpha -a}},\alpha >a$$
(56)$$f_{3}\left(\tilde{X}\right)=C_{3}{\left|\det\left(A^{\frac{1}{2}}\tilde{X}B{\tilde{X}}^{*}A^{\frac{1}{2}}\right)\right|}^{\gamma }e^{-b\eta \mathrm{tr}(A^{\frac{1}{2}}\tilde{X}B{\tilde{X}}^{*}A^{\frac{1}{2}})},\alpha \to a$$
where in (54), $I-b\left(a-\alpha \right)A^{\frac{1}{2}}\tilde{X}B{\tilde{X}}^{*}A^{\frac{1}{2}}>O$, and the normalizing constants are the following:(57)$$C_{1}={\left|\det\left(A\right)\right|}^{q}{\left|\det\left(B\right)\right|}^{p}\frac{{\tilde{\Gamma }}_{p}\left(q\right)}{\pi^{pq}}{\left[b\left(a-\alpha \right)\right]}^{p(\gamma +q)}\frac{{\tilde{\Gamma }}_{p}\left(\frac{p+1}{2}+\frac{\eta }{a-\alpha }+\gamma +q\right)}{{\tilde{\Gamma }}_{p}\left(\gamma +q\right){\tilde{\Gamma }}_{p}\left(\frac{\eta }{\alpha -a}\right)}\;$$
for α<a,b>0,γ>0,η>0,A>O,B>O;
(58)$$C_{2}={\left|\det\left(A\right)\right|}^{q}{\left|\det\left(B\right)\right|}^{p}\frac{{\tilde{\Gamma }}_{p}\left(q\right)}{\pi^{pq}}{\left[b\left(\alpha -a\right)\right]}^{p(\gamma +q)}\frac{{\tilde{\Gamma }}_{p}\left(\frac{\eta }{\alpha -a}\right)}{{\tilde{\Gamma }}_{p}\left(\gamma +q\right){\tilde{\Gamma }}_{p}\left(\frac{\eta }{\alpha -a}-\left(\gamma +q\right)\right)}\;$$
for $\alpha >a,b>0,\gamma >0,\eta >0,A>O,B>O,\frac{\eta }{\alpha -a}-\left(\gamma +q\right)>0$ and
(59)$$C_{3}={\left|\det\left(A\right)\right|}^{q}{\left|\det\left(B\right)\right|}^{p}\frac{{\tilde{\Gamma }}_{p}\left(q\right)}{\pi^{pq}}{\left[b\eta \right]}^{p(\gamma +q)}\frac{1}{{\tilde{\Gamma }}_{p}\left(\gamma +q\right)}\;$$
for A>O,B>O,b>0,η>0,γ>0. Note that the model in (56) can be taken as the complex rectangular matrix-variate Maxwell–Botzmann and Rayleigh densities for γ=1 and $\gamma =\frac{1}{2}$, respectively. The corresponding real cases were given by [12,13]. The standard versions of complex rectangular matrix-variate Maxwell–Boltzmann and Rayleigh densities are available from (56) by considering the density of $Y=A^{\frac{1}{2}}\tilde{X}B^{\frac{1}{2}}$. Then, in the normalizing constant $C_{3}$, ${\left|\det\left(A\right)\right|}^{q}{\left|\det\left(B\right)\right|}^{p}$ will be absent. The standard density is the following:(60)$$f_{4}\left(\tilde{Y}\right)=\frac{{\tilde{\Gamma }}_{p}\left(q\right)}{\pi^{pq}}{\left[\rho \right]}^{p(\gamma +q)}\frac{1}{{\tilde{\Gamma }}_{p}\left(\gamma +q\right)}{\left|\tilde{Y}{\tilde{Y}}^{*}\right|}^{\gamma }e^{-\rho \mathrm{tr}(\tilde{Y}{\tilde{Y}}^{*})}\;$$
where we have taken bη=ρ for convenience. Then, for p=1,q=1, we obtain the complex scalar versions of the Maxwell–Boltzmann and Rayleigh densities from (61) as the following:(61)$$f_{5}\left(\tilde{x}\right)=\frac{1}{\pi }\frac{\rho^{\gamma +1}}{\Gamma (\gamma +1)}{\left[\tilde{x}{\tilde{x}}^{*}\right]}^{\gamma }e^{-\rho \tilde{x}{\tilde{x}}^{*}},\rho >0,\gamma >0.\;$$
For γ=1, we have the Maxwell–Boltzmann density in the complex scalar case, and for $\gamma =\frac{1}{2}$, we have the Rayleigh density in the complex scalar case. Note that ${\left[\tilde{x}{\tilde{x}}^{*}\right]}^{\frac{1}{2}}=\left|\tilde{x}\right|=$, which is the absolute value of the scalar complex variable $\tilde{x}$. If the domain must be confined to $ℜ(\tilde{x})>0$, then multiply (61) by $\frac{1}{2}$.

### 8.1. Arbitrary Moments

As in the real case, we can consider the *h*-th moment of the absolute value of the determinant $\left|\det(A^{\frac{1}{2}}\tilde{X}B{\tilde{X}}^{*}A^{\frac{1}{2}})\right|$ for arbitrary *h*. Then, from (54), we have the following: For α<a,
$$\begin{array}{cc} E[|\det\left(b\left(a-\alpha \right)A^{\frac{1}{2}}\tilde{X}B{\tilde{X}}^{*}A^{\frac{1}{2}}\right){|}^{h}] & =\frac{{\tilde{\Gamma }}_{p}\left(\gamma +q+h\right)}{{\tilde{\Gamma }}_{p}\left(\gamma +q\right)}\frac{{\tilde{\Gamma }}_{p}\left(\gamma +q+\frac{\eta }{a-\alpha }\right)}{{\tilde{\Gamma }}_{p}\left(\gamma +q+\frac{\eta }{a-\alpha }+h\right)}\\ \; & =\prod_{j=1}^{p}\frac{\Gamma (\gamma +q-(j-1)+h)}{\Gamma (\gamma +q-(j-1))}\frac{\Gamma (\gamma +q+\frac{\eta }{a-\alpha }-\left(j-1\right))}{\Gamma (\gamma +q+\frac{\eta }{a-\alpha }-\left(j-1\right)+h)}\\ \; & =E\left[u_{1}^{h}\right]E\left[u_{2}^{h}\right]...E\left[u_{p}^{h}\right] \end{array}$$
for α<a and ℜ(h+γ+q)>p−1, where $u_{1},...,u_{p}$ are mutually independently distributed real scalar type-1 beta random variables, with $u_{j}$ having the parameters $(\gamma +q-\left(j-1\right),\frac{\eta }{a-\alpha }),j=1,...,p$. Therefore, we have the structural representation
(62)$$|\det\left(b\left(a-\alpha \right)A^{\frac{1}{2}}\tilde{X}B{\tilde{X}}^{*}A^{\frac{1}{2}}\right)|=u_{1}...u_{p}\;$$
where $u_{1},...,u_{p}$ are as defined above and α<a. From (55), we have the following: $$\begin{array}{cc} E[|\det\left(b\left(\alpha -a\right)A^{\frac{1}{2}}\tilde{X}B{\tilde{X}}^{*}A^{\frac{1}{2}}\right){|}^{h}] & =\frac{{\tilde{\Gamma }}_{p}\left(\gamma +q+h\right)}{{\tilde{\Gamma }}_{p}\left(\gamma +q\right)}\frac{{\tilde{\Gamma }}_{p}\left(\frac{\eta }{\alpha -a}-\left(\gamma +q\right)-\left(j-1\right)\right)}{{\tilde{\Gamma }}_{p}\left(\frac{\eta }{\alpha -a}-\left(\gamma +q\right)-\left(j-1\right)-h\right)}\\ \; & =\prod_{j=1}^{p}\frac{\Gamma (\gamma +q-(j-1)+h)}{\Gamma (\gamma +q-(j-1))}\frac{\Gamma (\frac{\eta }{\alpha -a}-\left(\gamma +q\right)-\left(j-1\right))}{\Gamma (\frac{\eta }{\alpha -a}-\left(\gamma +q\right)-\left(j-1\right)-h)}\\ \; & =E\left[v_{1}^{h}\right]E\left[v_{2}^{h}\right]\ldots E\left[v_{p}^{h}\right] \end{array}$$
for $\alpha >a,ℜ\left(h+\gamma +q\right)>p-1,ℜ(h+\frac{\eta }{\alpha -a}-\left(\gamma +q\right))>p-1$, where $v_{1},\ldots ,v_{p}$ are mutually independently distributed real scalar type-2 beta random variables with the parameters $(\gamma +q-\left(j-1\right),\frac{\eta }{\alpha -a}-\left(\gamma +q\right)-\left(j-1\right)),j=1,\ldots ,p$, and we have the structural representation
(63)$$|\det\left(b\left(\alpha -a\right)A^{\frac{1}{2}}\tilde{X}B{\tilde{X}}^{*}A^{\frac{1}{2}}\right)|=v_{1}\ldots v_{p}\;$$
for α>a, where $v_{1},\ldots v_{p}$ are defined above. From (56), we have the following results:$$\begin{array}{cc} E[|\det\left(b\eta A^{\frac{1}{2}}\tilde{X}B{\tilde{X}}^{*}A^{\frac{1}{2}}\right){|}^{h}] & =\frac{{\tilde{\Gamma }}_{p}\left(\gamma +q+h\right)}{{\tilde{\Gamma }}_{p}\left(\gamma +q\right)}=\prod_{j=1}^{p}\frac{\Gamma (\gamma +q-(j-1)+h)}{\Gamma (\gamma +q-(j-1))}\\ \; & =E\left[w_{1}^{h}\right]E\left[w_{2}^{h}\right]\ldots E\left[w_{p}^{h}\right] \end{array}$$
where ℜ(h+γ+q)>p−1, $w_{1},\ldots ,w_{p}$ are mutually independently distributed real scalar gamma random variables with the parameters (γ+q−(j−1),1),j=1,…,p, and we have the structural representation
(64)$$|\det\left(b\eta A^{\frac{1}{2}}\tilde{X}B{\tilde{X}}^{*}A^{\frac{1}{2}}\right)|=w_{1}w_{2}\ldots w_{p}\;$$
where $w_{1},\ldots ,w_{p}$ are defined above. Note that if q≤p and $\tilde{X}$ is of rank *q*, then we can obtain results for $B^{\frac{1}{2}}{\tilde{X}}^{*}A\tilde{X}B^{\frac{1}{2}}$ that are parallel to those in Section 8 and Section 8.1 by interchanging *p* with *q* and *A* with *B*.

## 9. Concluding Remarks

In this paper, it is shown that a large number of statistical densities belonging to the pathway family [16] of densities in the scalar, vector, and matrix-variate cases in the real and complex domains can be obtained by optimizing a certain entropy measure. The calculus of variation technique was used for the optimization. The notations were simplified and made consistent in order to avoid having too many symbols to denote different types of variables. Mathematical variables and random variables are treated in the same way to avoid the double notations that are usually used to denote random variables and the resulting confusions.

## Data Availability

Not applicable.

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
