# Peer review of "Entropy Optimization, Maxwell–Boltzmann, and Rayleigh Distributions"

_entropy, 2021, doi:10.3390/e23060754_

Round 1

Reviewer 1 Report

This is a purely mathematical paper without any

discernible physical or engineering application.

No conclusions are drawn. 

I would dare to assert that 99% of Entropy's readership will not

find this paper useful or interesting.

I recommend rejection.

Author Response

We are grateful to the Referee for comments and have improved the article accordingly.

Reviewer 2 Report

These are interesting results that can be derived from a particular form of entropy formulation. It would be useful to verify if Bose-Einstein and Fermi-Dirac distributions can be derived in a similar fashion.

I recommend the publication of the draft.

Author Response

We are grateful for the Referee's comments and have improved the article accordingly. Please see the attachment.

Reviewer 3 Report

Please find the file attached.

Author Response

We are grateful for the Referee's comments and have improved accordingly.

Round 2

Reviewer 1 Report

The paper can now be accepted.

Reviewer 3 Report

The authors supplemented the article with several inserts describing the meaning of their research. This slightly improved the quality of the article. I think the article can be published keeping in mind that the journal Entropy is not a purely mathematical journal.